# The Insulin Receptor Substrate 2 Mediates the Action of Insulin on HeLa Cell Migration via the PI3K/Akt Signaling Pathway

**Anabel Martínez Báez** [1]**, Ivone Castro Romero** [2]**, Lilia Chihu Amparan** [1]**, Jose Ramos Castañeda** [3]
**and Guadalupe Ayala** [1,*]

[1] Infection Disease Research Center, National Institute of Public Health, Cuernavaca 62100, Mexico
[2] Subdirectorate of Training and Medical Update, Secretary of Health, Mexico City 06900, Mexico
[3] Health Sciences School, Universidad Anahuac, Naucalpan 52786, Mexico
**\*** Correspondence: gayala@insp.mx

**Abstract:** Insulin signaling plays an important role in the development and progression of cancer since it is involved in proliferation and migration processes. It has been shown that the A isoform of the insulin receptor (IR-A) is often overexpressed, and its stimulation induces changes in the expression of the insulin receptor substrates (IRS-1 and IRS-2), which are expressed differently in the different types of cancer. We study the participation of the insulin substrates IRS-1 and IRS-2 in the insulin signaling pathway in response to insulin and their involvement in the proliferation and migration of the cervical cancer cell line. Our results showed that under basal conditions, the IR-A isoform was predominantly expressed. Stimulation of HeLa cells with 50 nM insulin led to the phosphorylation of IR-A, showing a statistically significant increase at 30 min ($p \leq 0.05$). Stimulation of HeLa cells with insulin induces PI3K and AKT phosphorylation through the activation of IRS2, but not IRS1. While PI3K reached the highest level at 30 min after treatment ($p \leq 0.05$), AKT had the highest levels from 15 min ($p \leq 0.05$) and remained constant for 6 h. ERK1 and ERK2 expression was also observed, but only ERK2 was phosphorylated in a time-dependent manner, reaching a maximum peak 5 min after insulin stimulation. Although no effect on cell proliferation was observed, insulin stimulation of HeLa cells markedly promoted cell migration.

**Keywords:** insulin receptor; IRS1; IRS2; PI3K/Akt; cell migration; cervical cancer

## 1. Introduction

Insulin plays an important role in the development and progression of cancer because it is involved in the processes of cell growth and proliferation due to its stimulatory effects on DNA synthesis in various tissues [1]. Insulin activates a tyrosine kinase receptor, the insulin receptor (IR), which undergoes autophosphorylation and phosphorylates endogenous substrates. Two different isoforms of the IR are generated by alternative splicing, IR-A and IR-B, which differ by the absence (IR-A) or presence (IR-B) of a 12-amino acid insert encoded by exon 11. IR-B is mainly expressed in the major insulin target tissues, whereas IR-A is predominantly expressed in the embryo and fetal tissues, central nervous system (CNS), hematopoietic cells, and several types of cancer cells [2]. When the IR is stimulated, the first proteins activated are adapter proteins, known as insulin receptor substrates (IRS). IRS-1 and IRS-2 are widely expressed in humans and are, therefore, the most studied proteins in the family [3]. There is a relationship between IRS-1 and 2 and various types of cancer, such as breast [4–8], lung [9], prostate [10], hepatocarcinoma [11–13], neuroblastoma [14,15], head and neck [16], colorectal [17,18], esophageal squamous cell carcinoma [19], non-small cell lung cancer [20], and glioblastoma multiforme [21]. It has been observed that the expression and function of the IRS may vary in the different types of cancer. For example, IRS-1 has been associated with proliferation, growth, and anti-apoptosis, whereas IRS-2 has been linked to metastasis, motility, and invasion [22–26]. IRS proteins participate in canonical

pathways, the phosphorylation of which is induced by the insulin receptor. The IR activates two main signaling pathways: the insulin receptor substrate/phosphatidyl inositol 3-kinase pathway (IRS/PI3-K) and the Ras/mitogen-activated protein kinase (MAPK) pathway. Both pathways regulate most of the effects of insulin, those associated with the regulation of energy metabolism, gene expression, and mitogenic effects [27].

The relationship between the expression levels of IRS-1 and IRS-2 and the activation of insulin signaling pathways has been poorly studied in cervical cancer cells. Although it has already been shown that SiHa cells (HPV16 +) express both IR-A and IR-B, only the activation of IR-A was related to the activation of Akt and ERK1/2. Akt and ERK1/2 participate in the phosphatidylinositol 3-kinase (PI3K) and MAPK pathways, respectively [28]. However, the roles of adaptor proteins IRS1 and IRS2 have not been investigated in this type of cancer.

The objective of this study was to investigate which isoform of the insulin receptor is expressed in the HeLa cervical cancer cell line and to analyze the role of IRS-1 and IRS-2 in the signaling pathway of the insulin receptor and in regulating the proliferation and migration of HeLa cells (HPV+).

## 2. Materials and Methods

### 2.1. Chemicals and Reagents

Recombinant human insulin was obtained from Sigma (St. Louis, MO, USA). MTS reagent was from PROMEGA (Wisconsin, WI, USA). TRIzol reagent, DNase I, EDTA, oligo dT, RNAsin, RT reaction buffer, DTT, dNTP's, reverse transcriptase, Taq DNA polymerase, and $MgCl_2$ were purchased from Invitrogen (Waltham, MA, USA). Protease and phosphatase inhibitor cocktail were obtained from Sigma. The 2D Quant commercial kit was from GE Healthcare Life Sciences (Chicago, IL, USA). SuperSignal™ West Femto Maximum Sensitivity Substrate was from Thermo Scientific (Waltham, MA, USA).

### 2.2. Cell Isolation and Culture

The human cervical cancer HeLa cell line was purchased from ATCC (Rockville, MD, USA), and human mammary epithelial MCF7 and human breast adenocarcinoma MDA-MB-231 cell lines were donated by Dra. Elizabeth Langley (National Cancer Institute, Mexico City, Mexico). Dulbecco's modified Eagle's medium (DMEM) and DMEM/F12 culture media were purchased from GIBCO. Heat-inactivated fetal bovine serum (FBS) and penicillin-streptomycin were obtained from GIBCO BRL (Carlsbad, CA, USA).

HeLa cells were cultured and maintained in DMEM, and MCF7 and MDA-MB-231 cell lines were cultured and maintained in DMEM/F12 supplemented with 10% FBS and antibiotics (penicillin/streptomycin 100 μg/mL) at 37 °C in a humidified atmosphere of 95% air and 5% $CO_2$. Cell viability was determined by Trypan blue dye exclusion method. The cell lines were seeded under sterile conditions at different densities. All cell lines were serum starved for 12 h prior to each experiment; cells were treated with 50 nM insulin (recombinant human insulin was purchased from Sigma, St. Louis, MO, USA) for indicated times.

### 2.3. Cell Proliferation Assays by MTS

The MTS assay was used to assess cell proliferation and cell viability. HeLa cells ($5 \times 10^3$ cells/well) were seeded in 96-well flat-bottomed tissue culture plates in three replicates, and incubated and supplemented with DMEM (low concentration of glucose 1 g/L) for 24 h. Next, the cells were washed once with 1X phosphate buffered saline (PBS) (137 mM NaCl, 2.7 mM KCl, 10 mM $Na_2HPO_4$, 2 mM $KH_2PO_4$). The 96-well tissue culture dishes were serum starved for 2 h. Cell proliferation was stimulated with 10, 50, or 100 nM insulin, and cell viability and proliferation were evaluated at 24, 48, and 72 h post-treatment. After the stimulation with insulin at different concentrations for the specified time, the medium was replaced with 2 mL of DMEM fresh medium supplemented with 0.25 mg/mL MTS reagent [3-(4,5-dimethylthiazol-2-yl)-5-(3-carboxymethoxyphenyl)-2-(4-

sulfophenyl)-2H-tetrazolium, inner salt] per well, and cells were incubated for 4 h at 37 °C. Then, 3-(4,5-dimethylthiazol-2-yl)-2,5-diphenyltetrazolium bromide formazan crystals were quantified at 595 nm using an absorbance microplate reader (iMark Microplate Reader, Bio-Rad, Hercules, CA, USA). All experiments were performed in three independent experiments in triplicate.

*2.4. RT-PCR*

The mRNA levels of IR, IRS1, IRS2, and GAPDH were detected using RT-PCR. Total RNA was isolated from the cells using TRIzol reagent (Invitrogen, Waltham, MA, USA). The total RNA (1.5 µg) was used for cDNA synthesis. Briefly, RNA was incubated with 1µL DNase for 15 min at room temperature, and the reaction was stopped by adding 1 µL EDTA (25 mM). The tubes were boiled at 65 °C for 10 min. Next, 1 µL of oligo dT (thymidine) (0.5 µg/µL) was added to each sample, incubated at 70 °C for 10 min, and placed on ice. For reverse transcription, a reaction mixture solution (0.5 µL of RNAsin (40 U/µL), 4.0 µL of RT reaction buffer (5X), 2.0 µL of DTT (0.1 M), 1.0 µL of dNTP's (10 mM), and 0.5 µL of reverse transcriptase (200 U/µL) was added to each tube and incubated at 37 °C (1 h) and 70 °C (15 min). Finally, tubes were placed on ice or stored at −20 °C until use. PCR was performed using the Taq PCR Master Mix kit (Invitrogen). PCR profiles for each primer pair were initially standardized over a series of cycles to ensure that all experimental reactions were performed within the linear range. The oligonucleotide primer sequences are listed in Table 1. The PCR products were analyzed by electrophoresis on 1.5% agarose gels.

**Table 1.** Oligonucleotides used for PCR.

| Target | Primer Sequence 5′-3′ | Position | Size (bp) |
|--------|----------------------|----------|-----------|
| IR | 5′-AACCAGAGTGAGTATGAGGAT-3′<br>5′-CCGTTCCAGAGCGAAGTGCTT-3′ | nt 2201-2221<br>nt 2780-2800 | Isoform B: 636<br>Isoform A: 600 |
| IRS1 | 5′-TCCACTGTGACACCAGAATAAT-3<br>5′-CGCCAACATTGTTCATTCCAA-3′ | nt 4011-4032<br>nt 4753-4773 | 763 |
| IRS2 | 5′-TAGGCATCAATGGGTGGTATTT-3′<br>5′-CTACGGATAGAGGGCGAGTTA-3′ | nt 6358-6380<br>nt 6453-6474 | 116 |
| GAPDH | 5′-ACCACAGTCCATGCCATCAC-3′<br>5′-TCCACCACCCTGTTGCTGTA-3′ | nt 602-621<br>nt 1031-1053 | 451 |

*2.5. Immunoprecipitation and Western Blot*

For protein analysis, cells were washed once with cold phosphate-buffered saline (PBS), lysed with RIPA buffer (50 mM Tris-HCl pH 7.4, 150 mM NaCl, 1 mM EDTA, 0.5% sodium deoxycholate, 0.1% SDS, 1% Nonidet) plus a protease and phosphatase inhibitor cocktail, and then boiled for 5 min at 95–100 °C. For protein quantification, we used a 2D Quant commercial kit (GE Healthcare Life Sciences, Chicago, IL, USA). From whole cell lysates, 40 µg of protein per lane was separated by SDS-PAGE and assayed by immunoblotting using specific antibodies for proteins of the IR signaling pathway, whereas 1.5–2 mg of protein was used for immunoprecipitation (IP) of IRS-2. Detailed information on the primary and secondary antibodies are given in Table 2. Proteins were detected by chemiluminescence using the commercial kit SuperSignal™ West Femto Maximum Sensitivity Substrate from Thermo Scientific (Waltham, MA, USA) using a C-Digit Blot Scanner (LI-COR Biosciences). β-Actin was used as a control to normalize the values of proteins of interest obtained by densitometry. Densitometric analysis was performed using ImageJ 1.47 software (National Institutes of Health, Bethesda, MD, USA).

**Table 2.** List of primary and secondary antibodies.

| Primary Antibody | Epitope/ Specificity | Host Species | Type | Dilution | Source |
|---|---|---|---|---|---|
| Insulin Receptor β (C19) | C-terminus/IgG | Rabbit | Polyclonal | 1:500 | Santa Cruz SC-711 |
| IRS1 (C-20) | C-terminus/IgG | Rabbit | Polyclonal | 1:500 | Santa Cruz SC-559 |
| IRS2 (H-205) | aa 926-1130/IgG | Rabbit | Polyclonal | 1:500 | Santa Cruz SC-8299 |
| PI 3-kinase (Z-8) | p85$\alpha$/IgG | Rabbit | Polyclonal | 1:500 | Santa Cruz SC-423 |
| ERK 1 (C-16) | C-terminus/IgG | Rabbit | Polyclonal | 1:500 | Santa Cruz SC-93 |
| Akt1 (B1) | aa 345-480/IgG$_1$ | Mouse | Monoclonal | 1:500 | Santa Cruz SC-5298 |
| Phospho insulin receptor | Phospho Tyr/1361/IgG | Rabbit | Polyclonal | 1:500 | Abcam ab60946 |
| Phospho-IRS1 | Phospho Tyr/632/ IgG | Rabbit | Monoclonal | 1:500 | Abcam ab109543 |
| Anti- phospho Tyrosine | Tyrosine-phosphorylated proteins/IgG2b | Mouse | Monoclonal | 1:500 | Millipore 05-947 |
| Phospho-Akt1 | Phospho Thr 308/IgG | Rabbit | Monoclonal | 1:500 | Millipore 05-802R |
| Phospho-Erk1/2 | Phospho Thr 202/Tyr 204, Thr 185/Tyr 187/IgG | Rabbit | Monoclonal | 1:500 | Millipore 05-797R |
| Phospho-PI3K p85 | p85 Phospho Tyr 467, Tyr 199/IgG | Rabbit | Polyclonal | 1:500 | GeneTex GTX132597 |
| β-actin (4E8H3) | IgG$_1$ | Mouse | Monoclonal | 1:500 | [29] |
| **Secondary antibody** | **Epitope Specificity** | **Host species** | **Type** | **Dilution** | **Source** |
| Rabbit anti-goat IgG HRP-coupled | | Rabbit | Polyclonal | 1:5000 | Santa Cruz SC-2768 |
| Mouse IgG HRP-coupled | | Mouse | Polyclonal | 1:5000 | GeneTex GTX213111-01 |

*2.6. Cell Migration Assays*

HeLa ($5 \times 10^5$/well) cells were seeded into cell-adherent 6-well plates and incubated for 24 h to form a monolayer confluence. Monolayers were washed twice with 1X PBS and incubated for 24 h in serum-free DMEM to establish the quiescence of cells, and then plates were incubated for 2 h with mitomycin C (12.5 μg) to eliminate the proliferative effect. For wound-healing assay, monolayers were vertically scratched using a p200 pipette tip after cells reached a confluency of 90–95%, and later washed to eliminate detached cells. A control photographic image was taken using a Nikon Eclipse TS 100 (40×) with a camera attachment. Subsequently, fresh culture serum-free medium containing different insulin concentrations (10 nM, 50 nM, and 100 nM) was added to each well, and the plates were incubated for 24 h. A second photographic image was taken for each condition. The rate of cell migration was measured as the percentage of wound area occupied by cells compared with the initial wound area using TScratch Software [30].

### 2.7. Statistical Analysis

The differences between treatment groups were analyzed using ANOVA, and statistical significance was determined using Tukey's HSD test. In all cases, statistical significance was set at $p < 0.05$. SPSS software (IBM, Armonk, NY, USA) was used for the statistical analysis.

## 3. Results

### 3.1. Insulin Receptor (IR) Isoforms Are Differentially Expressed in HeLa Cells

Figure 1 shows that the HeLa cells predominantly express IR-A (600 bp), with only a slight expression of IR-B (636 bp), whereas in the MCF-7 cell lines, similar amounts of both isoforms were expressed. Amplification of IRS-1 (763 bp) and IRS-2 (116 bp) fragments was observed at nearly equal levels in the two cell lines.

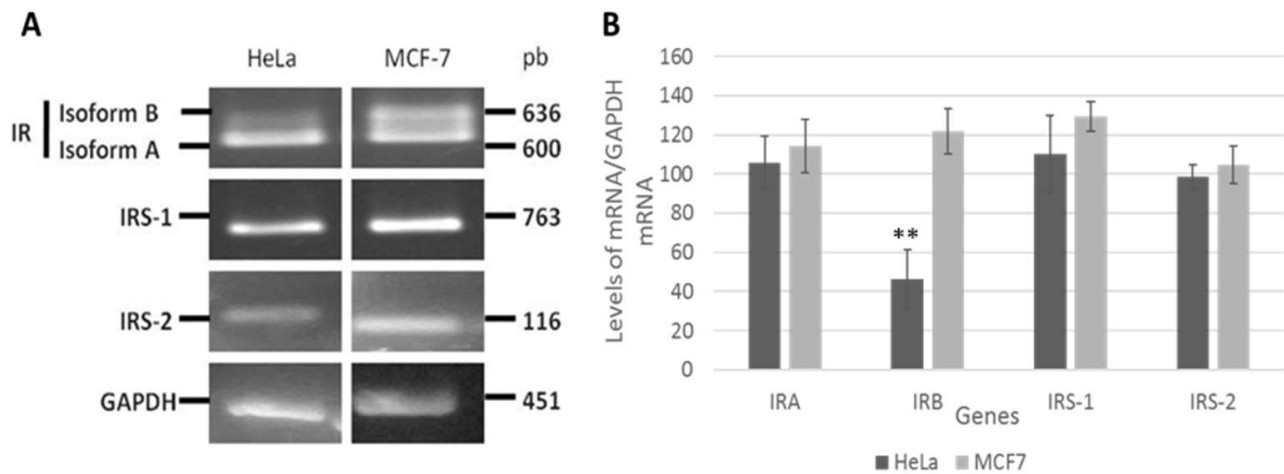

**Figure 1.** Insulin receptor (IR) isoforms are differentially expressed in HeLa cells. The mRNA levels of IR isoforms and IRS1/2 genes were analyzed in HeLa and MCF7 cell lines. Total RNA was purified, and mRNA levels were analyzed by RT-PCR with specific primers. (**A**) The 600 and 636 bp fragments correspond to isoforms A and B of the insulin receptor, respectively. Amplified fragments of 763 and 116 bp correspond to IRS-1 and IRS-2, respectively. The amplified fragment of 451bp corresponds to GAPDH (control). (**B**) Densitometric analysis of IRA, IRB, IRS1, IRS2, and GAPDH mRNA levels. The graph represents the mean $\pm$ SEM of three independent experiments ($n = 3$). ** $p < 0.01$ compared to IRA.

### 3.2. Effect of Insulin Treatment on Cell Proliferation

Cell proliferation is one of the main deregulated processes in cancer. We studied HeLa cell proliferation in response to insulin treatment. We used an MTS assay, a colorimetric method used to quantify viable cells based on the reduction of MTS to formazan by NAD-dependent dehydrogenase enzymes in metabolically active cells. Formazan was quantified by measuring the absorbance at a wavelength of 490 nm. As shown in Figure 2, we observed the proliferation of HeLa cells in response to different doses of insulin (10, 50, and 100 nM) and at different stimulation times (24, 48, and 72 h). The control group consisted of the unstimulated cells. There was a slight tendency for proliferation to increase with the 50 and 100 nM doses after 24 h of stimulation; however, the differences were not statistically significant.

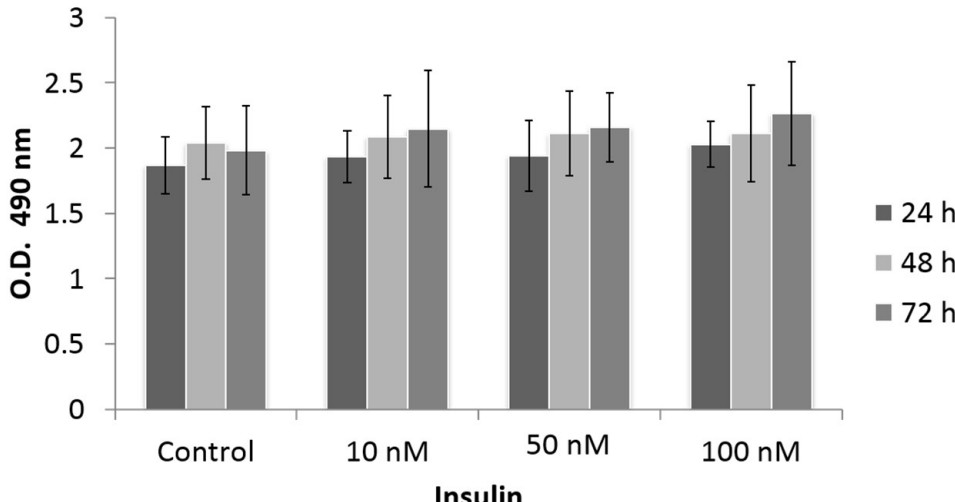

**Figure 2.** Effect of insulin on the proliferation of HeLa cells with MTS assay. HeLa cells were treated with different insulin doses at different times of stimulation. The histograms represent the mean value ± standard error of the mean (SEM) of optic density values. The control group is cells without treatment. The graphs represent the mean ± SEM of three independent experiments ($n = 6$).

### 3.3. Insulin Activates IR and IRS-2 but Not IRS-1 in HeLa Cells

We evaluated the ability of insulin to activate the insulin receptor and the IRS-1/2 substrates. As shown in Figure 3A, the expression of the non-phosphorylated β subunit of the insulin receptor did not change during the different incubation periods. The phosphorylated form showed a significant time-dependent increase under stimulation with 50 nM insulin ($p \leq 0.05$), reaching a maximum peak at 30 min. By analyzing the signaling pathways downstream of the IR, we found that insulin (50 nM) was able to stimulate IRS-2 tyrosine phosphorylation in the HeLa cells at different times (Figure 3B). After 15 min of stimulation, phosphorylated IRS-2 increased with respect to the control, reaching a peak at 30 min ($p \leq 0.05$). Interestingly, we did not observe IRS-1 phosphorylation in response to insulin treatment; however, the total protein levels did not change in the HeLa cells. There was phosphorylation of IRS-1 in the MCF7 cells (positive control) but no phosphorylation of IRS-1 in the MDA-MB-231 cells (negative control) (Figure 3C). These data suggest that, in this cell context, only the IRS2 pathway is activated, and IRS1 is not activated in response to insulin treatment.

### 3.4. PI3K/Akt1 Pathway Is Up-Regulated by Insulin in HeLa Cells

Next, we analyzed the signaling pathways downstream of IRSs. Two signaling pathways may be activated in response to insulin, the PI3K and MAPK cascades. The activation of the PI3K pathway was measured by PI3K and Akt1 phosphorylation, and the MAPK pathway was measured by Erk1/2 phosphorylation. Figure 4 shows the phosphorylation of PI3K and Akt1 after stimulation of the HeLa cells with 50 nM insulin. After insulin stimulation, the total protein content did not increase over time. The phosphorylated form of PI3K increased over time and was higher 30 min after insulin stimulation ($p \leq 0.05$) (Figure 4A). In Figure 4B, we show that insulin treatment increased the expression of the total AKT protein over time, with AKT reaching its highest expression 30 min after stimulation. However, phosphorylated p-Akt1 predominated at 15 min ($p \leq 0.05$) and remained constant until 6 h after insulin stimulation.

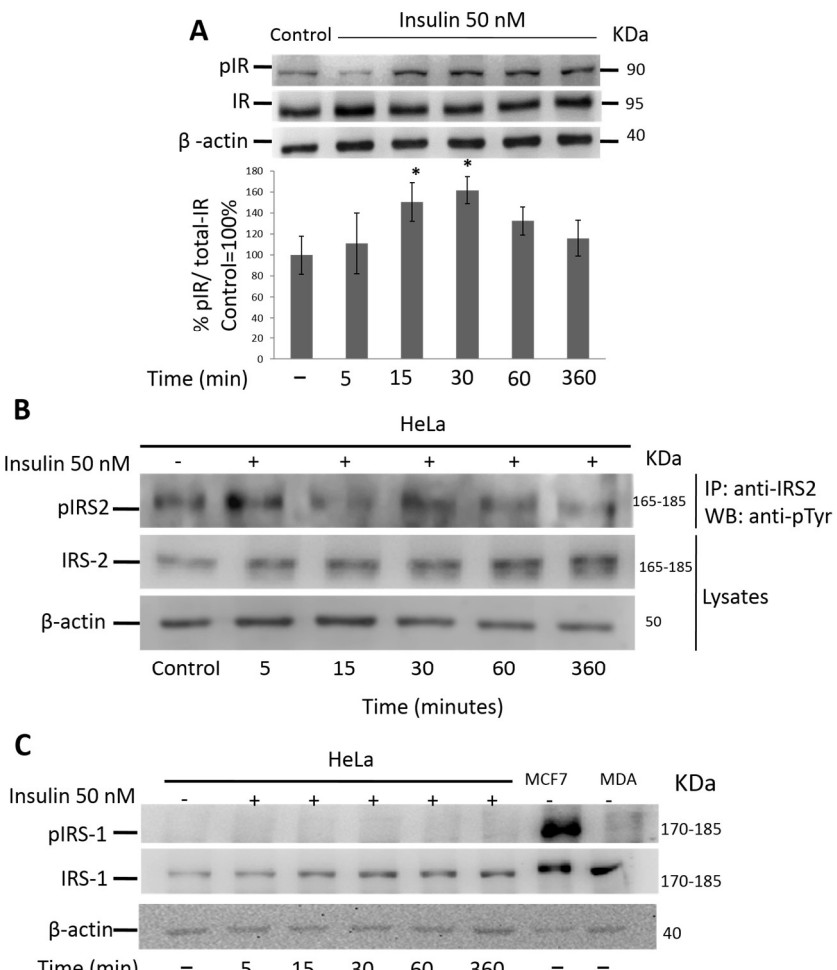

**Figure 3.** Insulin induces the phosphorylation of IR and IRS2 but not IRS1 in HeLa cells. Protein extracts from HeLa cells stimulated with 50 nM insulin were used to evaluate IR, IRS1, phospho-IRS1, IRS2, and phospho IRS2 by WB. (**A**) Densitometric analysis of phospho-IR and actin protein levels. The first bar is the control group without treatment. One-way ANOVA was performed, followed by the Tukey post hoc test to compare the treated groups against the control group (100%). * $p < 0.05$. All experiments have been performed in three independent experiments in triplicate, and experimental data were expressed as mean ± standard deviation (SD). (**B**) Cells were lysed with RIPA. IRS2 protein was immunoprecipitated, and immunoblot analyses were performed to identify the indicated proteins from IP or cell lysates. (**C**) HeLa cells were stimulated in the absence (control) or presence of 50 nM insulin for the indicated time. MCF7 and MDA-MB-231 cells were used as positive and negative controls, respectively. IRS1 and phospho-IRS1 protein levels were detected by immunoblot analysis. β-Actin was used as a control for protein degradation.

### 3.5. MAPK Signaling Pathway Is Not Activated by Insulin Treatment in HeLa Cells

IRS activates the MAPK signaling cascade MAPK. To explore whether the mitogenic pathway is also activated in the HeLa cells in response to insulin we measured Erk1/2 phosphorylation. The data showed that the total ERK1/2 proteins and their phosphorylation did not increase over time (Figure 4C). These data suggest that insulin was not able to activate the MAPK signaling pathway in our cellular model.

### 3.6. Insulin Induces Migration of HeLa Cells

We investigated the effect of insulin on cell migration using a wound-healing assay at different insulin doses. Figure 5 shows that 50 and 100 nM insulin significantly increased HeLa cell migration compared with the group of cells that did not receive insulin treatment at 48 h. This effect was different in the case of the HaCaT cells (non-transformed cells),

where the percentage of the open area was lower in all insulin doses compared with the control group without treatment.

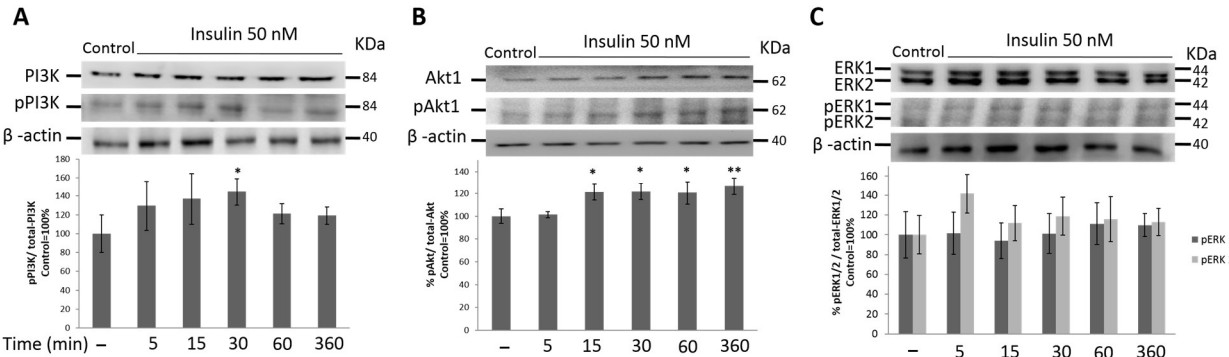

**Figure 4.** Cervical cancer cells express higher phosphorylation levels of PI3K/Akt1 than ERK1/2 in response to insulin treatment. (**A**) PI3K, phospho PI3K. (**B**) Akt1, phospho Akt1. (**C**) ERK1/2 and phospho ERK1/2 protein levels were analyzed by WB in whole-cell lysates from HeLa cells stimulated with 50 nM insulin. Data are representative of 3 independent experiments. Graphics show densitometric analysis of total levels and phosphorylated protein. The first bar is the control group without treatment. All experiments were performed in three independent experiments in triplicate, and experimental data were expressed as mean ± standard deviation (SD). * $p < 0.05$ and ** $p < 0.01$ compared to control (taken as 100%).

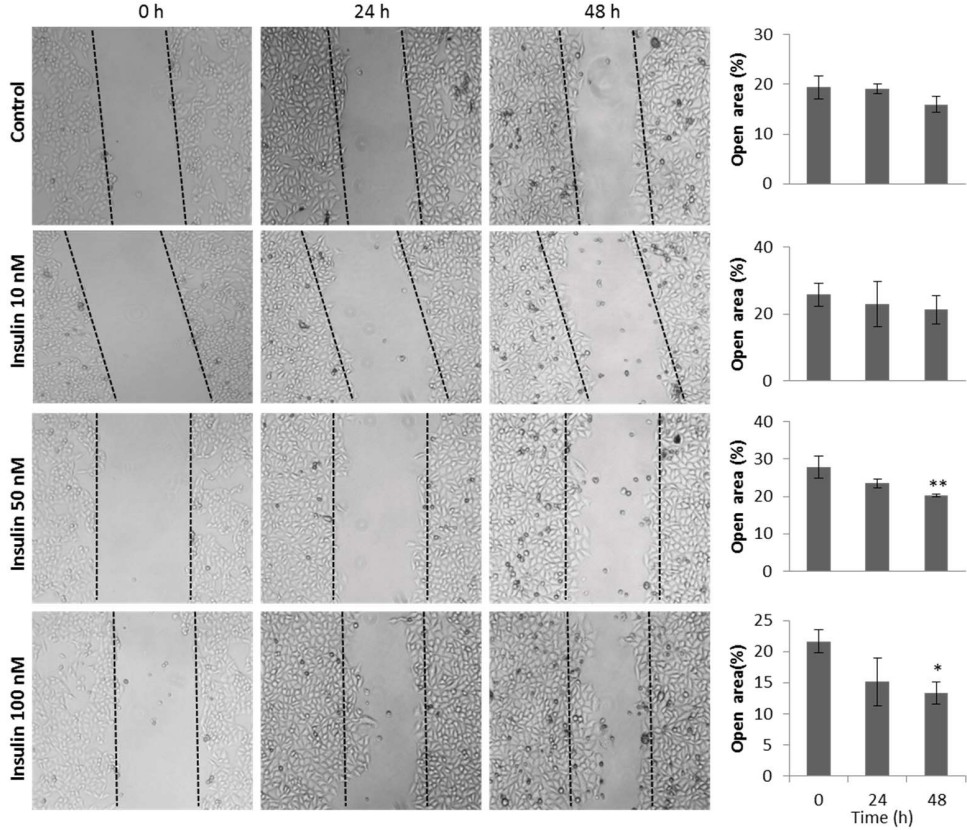

**Figure 5.** Insulin treatment increases migration of HeLa cells. HeLa cells were stimulated with different concentrations of insulin in the presence of mitomycin C and with DMEM medium without SFB; cells were treated with mitomycin C in the control group. Cell migration was evaluated by quantifying the reduction in the open area (a lower percentage of the open area means a higher percentage of cell migration). The histograms represent the mean value ± standard error of the mean (SEM) of the percentage of the open area of three independent experiments in triplicate. One-way ANOVA was performed, followed by the Tukey post hoc test to compare the treated groups against the control group. * $p < 0.05$; ** $p < 0.01$.

## 4. Discussion

Several studies have suggested that the insulin signaling pathway plays an important role in the development and progression of cancer, as it is involved in cell growth and proliferation processes due to its capacity to stimulate DNA synthesis in various tissues [1]. Several epidemiological studies and experimental models of insulin resistance and hyper-insulinemia have shown a correlation between insulin levels and cancer development. In cancer patients affected by insulin resistance, the increase in circulating levels of insulin is combined with the frequent overexpression of the insulin receptor in cancer cells, resulting in the abnormal stimulation of non-metabolic effects of the IR, such as cell survival, proliferation, and migration [31]. Alterations in insulin signal transduction increase the risk of cancer development.

Additionally, different groups have suggested that IRS1 and IRS2 are involved in cell growth, proliferation, migration, and metastasis [3]. Many studies have focused on the increased expression level or activity of IRSs in different human cancers, including breast, lung, and colorectal cancer, and have correlated these with poor prognosis, potentially defining IRSs as oncogenic proteins [32].

Notably, there is very little information in the literature related to the role of the insulin signaling pathway in the carcinogenesis of cervical cancer. It has been reported that progesterone upregulates IRS-2 expression, altering the levels of IRS-1 and IRS-2 in HeLa cells expressing progesterone receptors [33]; however, very little is known about the role of the insulin signaling pathway in cell proliferation and migration in cervical cancer. Additionally, it has been shown previously that SiHa cells (HPV16 +) express both IR-A and IR-B [28]. This suggests that the insulin signaling cascade is involved in the growth and proliferation of cervical cancer cells.

In this study, we investigated the activation of the insulin signaling pathway associated with insulin treatment in the HeLa cell line. Initially, we characterized the HeLa cell line based on the expression of the IR and two substrates, IRS-I and IRS-2. As shown in Figure 1, the HeLa cells predominantly expressed IR-A (600 bp) under basal culture conditions. In contrast, the MCF7 cells (positive control) expressed both IR-A (600 bp) and IR-B (630 bp). Similarly, Serrano et al. [28] showed that C33-A cells only express IR-A, whereas the SiHa cervical cell line expresses both isoforms. IR-A is predominantly expressed in fetal tissues; this isoform is less expressed in differentiated tissues from adults, such as the liver, muscle, and adipose tissue, classic targets of the metabolic effects of insulin, where IR-B expression predominates. However, IR-A continues to be expressed in some adult tissues, which are not the typical targets of insulin. For example, IR-A is often overexpressed in breast cancer [34], thyroid cancer, colon cells [35], and hepatocellular carcinoma [36]. The IR-A was more potent than the IR-B in mediating cell migration, invasion, and in vivo tumor growth in triple-negative breast cancer [37]. Although the precise biological roles of the two IR isoforms are unknown, it has been suggested that cancer cells preferentially express isoform A because they dedifferentiate and recover a 'fetal-like' phenotype [3,36].

We focused on studying the effect of different concentrations of human recombinant insulin (10, 50, and 100 nM) using an MTS assay to assess cell proliferation at 24, 48, and 72 h. Our results hinted at a minimal increase in the proliferation of HeLa cells treated with different insulin doses, but statistical analysis of the data showed no significant difference with respect to the control, suggesting that insulin does not affect the proliferation of HeLa cells. These results are similar to those reported by Serrano et al. [28] in the SiHa and C33-A cervical cancer cell lines, as they did not observe any effect on proliferation upon stimulation with IGF-I, IGF-II, or insulin in these cell lines. However, in thyroid cancer, insulin at supra-physiological concentrations promotes thyroid cell proliferation [38]. In addition, a previous study showed that astrocyte cell numbers increased in a dose-dependent manner upon insulin treatment [39]. This could indicate that the proliferative effect of insulin is tissue-specific and dependent on the insulin concentration. In addition, IRS1 is related to cell proliferation in cancer, and we did not observe the phosphorylation of IRS1 in this study.

Next, we found that insulin stimulated IR autophosphorylation, consistent with the presence of receptors in HeLa cells. Downstream of the insulin receptors, we observed the expression of IRS-1 and IRS-2. Surprisingly, only the phosphorylation of IRS-2 increased; in contrast, we did not observe the activation of IRS1. These findings suggest that in this cell line, IRS-2 is predominantly active. IRS-2 is generally related to processes such as metastasis, migration, and cell invasion in different types of cancer, while IRS-1 is related to proliferation. IRS2 is expressed at high levels in breast carcinoma cells of the basal-like/triple-negative breast cancer (TNBC) subtypes, and it regulates tumor cell migration, invasion, and glycolytic metabolism. The different functions of IRS1 and IRS2 in breast cancer are further evidenced by the fact that mouse mammary tumors lacking IRS2 have a significantly diminished ability to metastasize to the lungs, whereas tumors lacking IRS1 but expressing elevated IRS2 have enhanced metastatic potential [4,24,40,41]. In contrast, a recent study provided evidence that IRS1, rather than IRS2, is a dominant regulator of pancreatic alpha-cell function [42]. In breast cancer, IRS1 overexpression also promotes the growth and proliferation of BT 20 cells and induces the formation of larger tumors in vivo [43]. In lung cancer, tumors with low IRS-1 and high IRS-2 expression were associated with poor outcomes in adenocarcinoma and squamous cell carcinoma, indicating a potential role for IRS-2 in the aggressive behavior of non-small cell lung cancer [25]. These findings indicate that IRS1 and IRS2 play different roles depending on the cellular context; IRS2 is primarily responsible for cell motility and metastasis, whereas IRS1 is mainly important for cell proliferation [3].

IRS1- and IRS2-induced signaling is highly modulated during many cancer processes, such as cell motility, metastasis, and cell proliferation. Therefore, we focused on studying the molecular mechanisms involved in controlling the migration of HeLa cells after insulin treatment. In our model, we observed increased PI3K and Akt phosphorylation; however, we did not observe significant phosphorylation of ERK1/2. These data suggest that the PI3K pathway is activated mainly in response to insulin. Other studies have shown that in transgenic mice that do not express IRS-1, there is an increased function of IRS-2 and very high PI3K/Akt/mTor activity [44]. In addition, Hippo signaling interacts with AKT signaling by regulating IRS2 expression to prevent liver cancer progression [12]. However, in SiHa cells, a cell line transformed with HPV genotype 16, both the PI3K and MAPK pathways are activated in response to insulin and IGF-1 [28].

Carcinogenesis is complex. Normal cells undergo multiple genetic mutations before transformation to the complete neoplastic phenotype of growth, invasion, and metastasis. We investigated the effect of insulin on cell migration. Tumor cells are known to have accelerated metabolic rates and high glucose demand in a nutrient-poor environment [45,46]. The combination of these factors may result in a metabolic dependence on a continuous energy and nutrient supply for cells within the tumor mass [47]. We used a relatively low concentration of glucose (1000 mg/mL; 5.55 mM) in our experiments; according to Ishida et al. [48], a low glucose concentration increased the total migration length of HeLa cells and that HeLa cells under a glucose concentration gradient exhibit random motion rather than chemotaxis. However, the differences in migration of the HeLa cells they used are observed at a concentration of 0.7 mM; therefore, although we cannot rule out an effect on migration by the concentration of glucose used in our experiments, we consider that this does not essentially change the interpretation of the observations. As expected, our results showed a statistically significant increase in HeLa cell migration upon stimulation with insulin. This correlates with several reports on neuroblastoma and breast cancer, where the overexpression of IRS-2 promoted cell motility, invasion, and metastasis [40,44]. In addition, insulin promotes the migration of neural cells [49], thyroid cells [38], vascular smooth muscle cells [50], and advanced prostate cancer (PCa) [51]. Actually, the actions of insulin in PCa cells may be suppressed by inhibiting the downstream signaling molecules PI3K and ERK1/2 [51]. Interestingly, a recent study revealed that the ability of IRS2 to promote invasion is dependent upon upstream insulin-like growth factor 1 receptor (IGF-1R)/IR activation and the recruitment and activation of PI3K, which are functions shared

with IRS1. In addition, a 174-amino-acid region in the IRS2 C-terminal tail, which is not conserved in IRS1, is required for IRS2-mediated invasion [52].

## 5. Conclusions

There is a lack of definitive information on the role of insulin in cancer, and the situation is made more complex by the existence of two insulin receptor isoforms, IR-A and IR-B. We seek to address this void by examining insulin signal transduction in the cervical cancer cell line HeLa, which has not previously been examined.

The present study demonstrates that HPV-positive HeLa cells mainly express the IR-A isoform of the insulin receptor. Additionally, the insulin signaling pathway has been shown to be functionally active in these cells through the activation of the PI3K cascade via IRS2, thereby increasing cell migration. Further studies are necessary to clarify the roles of IR-A and IRS2 in metastatic processes and cancer cell progression.

**Author Contributions:** I.C.R. designed the project; A.M.B. and G.A. designed the experiments; A.M.B. led and performed the experiments; A.M.B., G.A. and L.C.A. searched and analyzed the literature, interpreted the results, and contributed to the discussion; A.M.B., G.A. and I.C.R. prepared and wrote the original draft; A.M.B., G.A. and L.C.A. wrote, reviewed, and edited; I.C.R. and G.A. acquired the funds. J.R.C. reviewed, contributed to the discussion, and edited the final version of the manuscript. All authors have read and agreed to the published version of the manuscript.

**Funding:** This research was funded by CONACyT project no. CB-2011-168375-B.

**Data Availability Statement:** The datasets used and/or analyzed during the current study are available from the corresponding author on reasonable request.

**Acknowledgments:** Anabel Martínez Baez is a doctoral student from Programa de Doctorado en Ciencias Biomédicas, Universidad Nacional Autónoma de México (UNAM) and received fellowship 420719 from CONACYT, México. We thank José Manuel Hernández (CINVESTAV, IPN, Mexico City, Mexico) for providing us with β-actin (4E8H3) antibody.

**Conflicts of Interest:** The authors state that they have no conflict of interest to declare.

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
