# Peer review of "The Insulin Receptor Substrate 2 Mediates the Action of Insulin on HeLa Cell Migration via the PI3K/Akt Signaling Pathway"

_cimb, doi:10.3390/cimb45030148_

Round 1

Reviewer 1 Report

The manuscript uses cervical cells lines (HeLa) to determine the involvement of IRS-1 and 2 in the insulin signaling mechanism and whether it plays a role in proliferation and migration of cancer cells. The authors show that IR-A is predominantly expressed and downstream signaling including PI3K, AKT and IRS2. They also show that is not mediated by IRS1. Authors also show that insulin promoted cell migration and not proliferation. Comments and suggestions are included as it appears in the text.

1.       Line 15, please change to cervical cell line. Only HeLa was tested.

2.       It is recommended that the antibodies be shown as a table and if possible with dilutions.

3.       Line 93 is it mM or nM. Throughout the text it says mM, if so, that is very high concentration. Please reconcile. Figures

4.       Figure 1 and 2 has no statistical analysis, so how is it decided that there is an increase or decrease or change?

5.       Table 1- where the TaqMan primers designed by the authors, is there a probe sequence?

6.       Line 134- Hacat was not described earlier.

7.       Line 156 was this statistically significant? What test was used? Please show significance.

8.       In general, how many times were these repeated- technical and biological repeats of the HeLa cells?

9.       Fig 3b-was IRS2 and b actin probed in the immunoprecipitated lysate or in total protein. The labeling is confusing in the figure. Please clarify

10.   Fig 3c- generally phospho bands come before total bands

11.   Fig 4- was band density calculated as pAKT/total AKT? If not was there no difference between total AKT across treatments? 

Reviewer 2 Report

The manuscript analyzed which isoform of the insulin receptor is expressed in the HeLa cell line. In addition, the role of IRS-1 and IRS-2 in the insulin receptor signaling pathway and the regulation of proliferation and migration of HeLa cells were investigated.

The work could be interesting, but it needs to be conceptualized better. Also, there are many inaccuracies and absence of information in the methodological part.

Comments

Check the accuracy of the sentences in the abstract (line 13)

Section "2. Materials and Methods" lacks information about many reagents and their origin. The authors should first of all add a new section  “2.1. Materials”

Line 57 - 258 etc....  The statement "cell lines for cervical cancer" is incorrect considering that the manuscript focuses only on the HeLa cell line

Section 2.2 -  I suggest preparing a table with the complete characteristics of the antibodies used (specificity, animal, source, dilution  etc…)

Section 2.3 of cell proliferation assays by MTS should be further specified. Also indicate the glucose concentration of the cell culture medium used. The role of glucose in cell migration needs to be discussed, particularly regarding its role the tumor microenvironment.

Section 2.5 - on RT-PCR much information is missing about reagents and methods

Check insulin concentrations (10 mM ??) in the text, e.g. line 141, in Fig. 2, line 301 etc....

The authors must show the reviewers, the original images of the different Western-blotting as they were performed in three independent experiments in triplicate.

Authors are requested to add more recent scientific references about the topic of the manuscript

The manuscript requires extensive revision of the text and affirmations.  The methodological part needs to be improved. Discussion of results must be improved.

Round 2

Reviewer 2 Report

Authors have taken in account the comments and the new revised  version  of the manuscript has been improved.

 The manuscript can been accepted in the present revised form.